# Ablation of CD8+ T cell recognition of an immunodominant epitope in SARS-CoV-2 Omicron variants BA.1, BA.2 and BA.3

Srividhya Swaminathan [1,2], Katie E. Lineburg[1], Archana Panikkar[1], Jyothy Raju[1], Lawton D. Murdolo [3], Christopher Szeto [3,4], Pauline Crooks[1], Laetitia Le Texier[1], Sweera Rehan[1], Michael J. Dewar-Oldis[3], Peter J. Barnard [3], George R. Ambalathingal[1], Michelle A. Neller[1], Kirsty R. Short [5], Stephanie Gras [3,4], Rajiv Khanna[1,2] & Corey Smith [1,2] ✉

The emergence of the SARS-CoV-2 Omicron variant has raised concerns of escape from vaccine-induced immunity. A number of studies have demonstrated a reduction in antibody-mediated neutralization of the Omicron variant in vaccinated individuals. Preliminary observations have suggested that T cells are less likely to be affected by changes in Omicron. However, the complexity of human leukocyte antigen genetics and its impact upon immunodominant T cell epitope selection suggests that the maintenance of T cell immunity may not be universal. In this study, we describe the impact that changes in Omicron BA.1, BA.2 and BA.3 have on recognition by spike-specific T cells. These T cells constitute the immunodominant CD8+ T cell response in HLA-A*29:02+ COVID-19 convalescent and vaccinated individuals; however, they fail to recognize the Omicron-encoded sequence. These observations demonstrate that in addition to evasion of antibody-mediated immunity, changes in Omicron variants can also lead to evasion of recognition by immunodominant T cell responses.

T cell immunity augments vaccine-mediated protection against SARS-CoV-2, particularly in the face of emerging variants such as Omicron[1,2], and is likely to be critically important in individuals who fail to generate efficient neutralizing antibody responses due to underlying disease[3,4]. Following the emergence of Omicron, studies have demonstrated that most T cell epitopes are conserved[5,6], and the majority of vaccine recipients maintain T cell immunity[7–9]. Nevertheless, evidence in convalescent individuals has demonstrated the capacity for variant strains to evade T cell responses[10]. In this study, we sought to investigate the impact of changes in Omicron on a uniquely immunodominant spike-encoded CD8+ T cell epitope.

## Results and discussion

Our group recruited 59 COVID-19 convalescent participants during the initial 2020 wave of the pandemic in Australia. All participants were typed for human leukocyte antigens (HLA, Supplementary Table 1) and SARS-CoV-2-specific T cells were expanded from peripheral blood mononuclear cells (PBMC) using SARS-CoV-2-encoded antigens[11]. In this analysis, we noted striking immunodominant CD8+ T cell responses to both spike (S-1 pool) and ORF3A in HLA-A*29:02+ participants (Fig. 1A, Supplementary Fig. 1). We detected a median S-1 pool-specific CD8+ T cell response of 44.6% in HLA-A*29:02+ participants compared to 0.85% in HLA-A*29:02− participants (Fig. 1B). The ORF3A-specific

[1]QIMR Berghofer Centre for Immunotherapy and Vaccine Development and Translational and Human Immunology Laboratory, Infection and Inflammation Program, QIMR Berghofer Medical Research Institute, Herston, QLD 4006, Australia. [2]Faculty of Medicine, The University of Queensland, Herston, QLD 4006, Australia. [3]Department of Biochemistry and Chemistry, La Trobe Institute for Molecular Science, La Trobe University, Melbourne, VIC 3086, Australia. [4]Department of Biochemistry and Molecular Biology, Biomedicine Discovery Institute, Monash University, Clayton, VIC 3800, Australia. [5]School of Chemistry and Molecular Biosciences, The University of Queensland, St Lucia, QLD 4072, Australia. ✉e-mail: corey.smith@qimrberghofer.edu.au

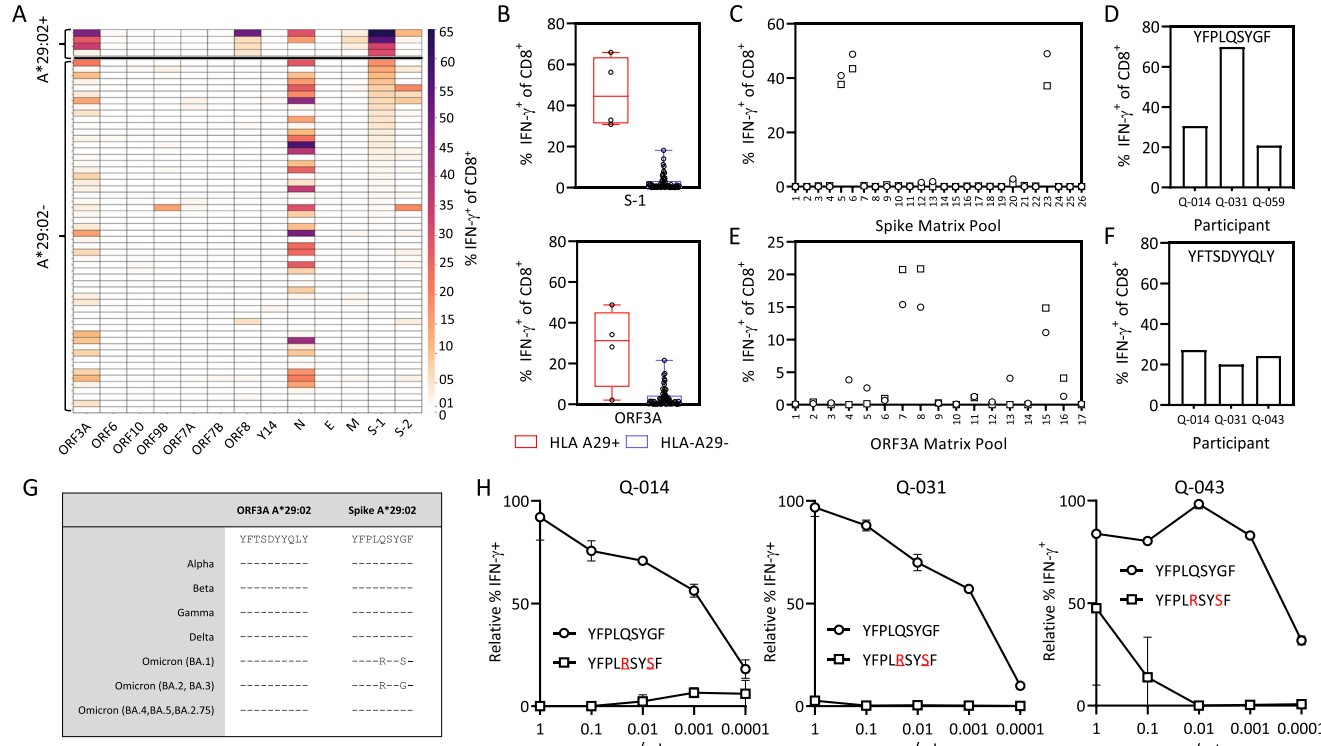

**Fig. 1 | Immunodominant HLA-A*29:02-restricted T cell immunity in COVID-19 convalescent participants.** SARS-CoV-2 specific T cells from COVID-19 convalescent participants were expanded from PBMC using SARS-CoV-2 encoded antigens. PBMCs were cultured for 2 weeks in the presence of IL-2 and assessed for IFN-g responses following recall with the specific antigen. **A** Heatmap of the magnitude of SARS-CoV-2 antigen-specific T cell responses in COVID-19-convalescent individuals (HLA-A*29:02⁺, n = 4; HLA-A*29:02⁻, n = 55). **B** Frequency of S-1- and ORF3A-specific IFN-γ-producing CD8⁺ T cells in HLA-A*29:02⁺ and HLA-A*29:02⁻ participants. Whiskers represent minimum to maximum values, boxes represent the 25th to 75th percentile, line represents the median. **C** Frequency of IFN-γ-producing CD8⁺ T cells in response to S-1 peptide matrix pools in HLA-A*29:02⁺ participants (n = 2). **D** Frequency of IFN-γ-producing CD8⁺ T cells in response to the YFPLQSYGF epitope in HLA-A*29:02⁺ participants. **E** Frequency of IFN-γ-producing CD8⁺ T cells in response to the ORF3A peptide matrix pools in HLA-A*29:02⁺ participants (n = 2). **F** Frequency of IFN-γ-producing CD8⁺ T cells in response to the YFTSDYYQLY epitope in HLA-A*29:02⁺ participants. **G** Prevalent HLA-A*29:02 epitope sequences in SARS-CoV-2 variants of concern. **H** Frequency of IFN-γ-producing CD8⁺ T cells in response to serial dilutions of YFPLQSYGF and YFPLRSYSF in HLA-A*29:02⁺ participants. Data represent mean and standard deviation of duplicates. Source data are provided as a source data file.

CD8⁺ T cell response was detected at a median of 31.2% in HLA-A*29:02⁺ participants compared to 1.47% in HLA-A*29:02⁻ participants. Using an overlapping peptide matrix, we identified that the spike-specific T cell response targeted spike matrix pools 5, 6 and 23 (Fig. 1C), corresponding to overlapping peptides GFNCYFPLQSYGFQP and YFPLQSYGFQPTNGV (Supplementary Fig. 2A). Peptide minimization and HLA restriction (Supplementary Fig. 2B) showed the immunodominant response to be encoded by a HLA-A*29:02-restricted epitope, YFPLQSYGF (Fig. 1D). The immunodominant response in matrix pools 7, 8 and 15 from ORF3a (Fig. 1E) corresponded to overlapping peptides VVLHSYFTSDYYQLY and SYFTSDYYQLYSTQL (Supplementary Fig. 2C), which contain the previously published HLA-A*29:02-restricted epitope YFTSDYYQLY (Fig. 1F)[12].

Analysis of the amino acid sequences of the HLA-A*29:02 epitopes in SARS-CoV-2 variants showed that while the ORF3A-encoded epitope was conserved, the Omicron BA.1 variant contained amino acid substitutions at positions 5 and 8 in the spike-encoded epitope (Fig. 1G). The substitution at position 5 was also evident in the Omicron BA.2 and BA.3 variants, whilst the Omicron BA.4, BA.5 and BA.2.75 variants had reverted to the wild-type sequence. Although not anchor residues for A*29:02, the substitutions, particularly the non-conserved glutamine (Q) to arginine (R) change at position 5, have the potential to significantly impact T cell receptor (TCR) engagement. To address this, we generated T cells specific for the YFPLQSYGF epitope from three COVID-19-convalescent participants and performed intracellular cytokine analysis using 10-fold serial peptide titration with both the cognate epitope and Omicron variant. Despite strong functional

avidity directed towards the cognate peptide (Supplementary Fig. 3A), we saw poor reactivity against the Omicron variant (Fig. 1H, Supplementary Fig. 3B). To assess the impact of single amino acid substitution at position 5, YFPL**R**SYGF and position 8, YFPLQSY**S**F, functional avidity assays with the single mutant peptides were performed. We saw poor reactivity against the both YFPL**R**SYGF and YFPLQSY**S**F single mutated peptide in Q031 (Supplementary Fig. 3C) and Q043 (Supplementary Fig. 3D).

We next assessed if individuals vaccinated against SARS-CoV-2 would show a similar pattern in their response to YFPLQSYGF. From a cohort of 104 participants, recruited following vaccination with either an adenovirus vector (Vaxzevria) or an mRNA (Cominarty) vaccine, we identified three participants with HLA-A*29:02 (Supplementary Table 2). PBMC from these individuals were isolated 28 days after their second vaccine dose, then stimulated with a spike peptide pool, the YFPLQSYGF epitope or the Omicron variant. We assessed expression of IFN-γ and TNF by CD8⁺ T cells using intracellular cytokine analysis to identify low-frequency T cell responses. Controls were unstimulated PBMC (no peptide) and PBMC incubated with a cell stimulation cocktail (CSC). Although low in abundance, IFN-γ⁺TNF⁺ CD8⁺ T cell responses against the spike overlapping peptide pools (0.006%, 0.009%, and 0.002% above no peptide) and the YFPLQSYGF epitope (0.046%, 0.012%, and 0.006% above no peptide) were detected in PBMC from all three individuals (Fig. 2A). Similar observations were evident following analysis of IFN-γ alone (Supplementary Fig. 4A). Conversely, CD8⁺ T cells capable of recognizing the Omicron variant were not detected. To validate these observations, we generated

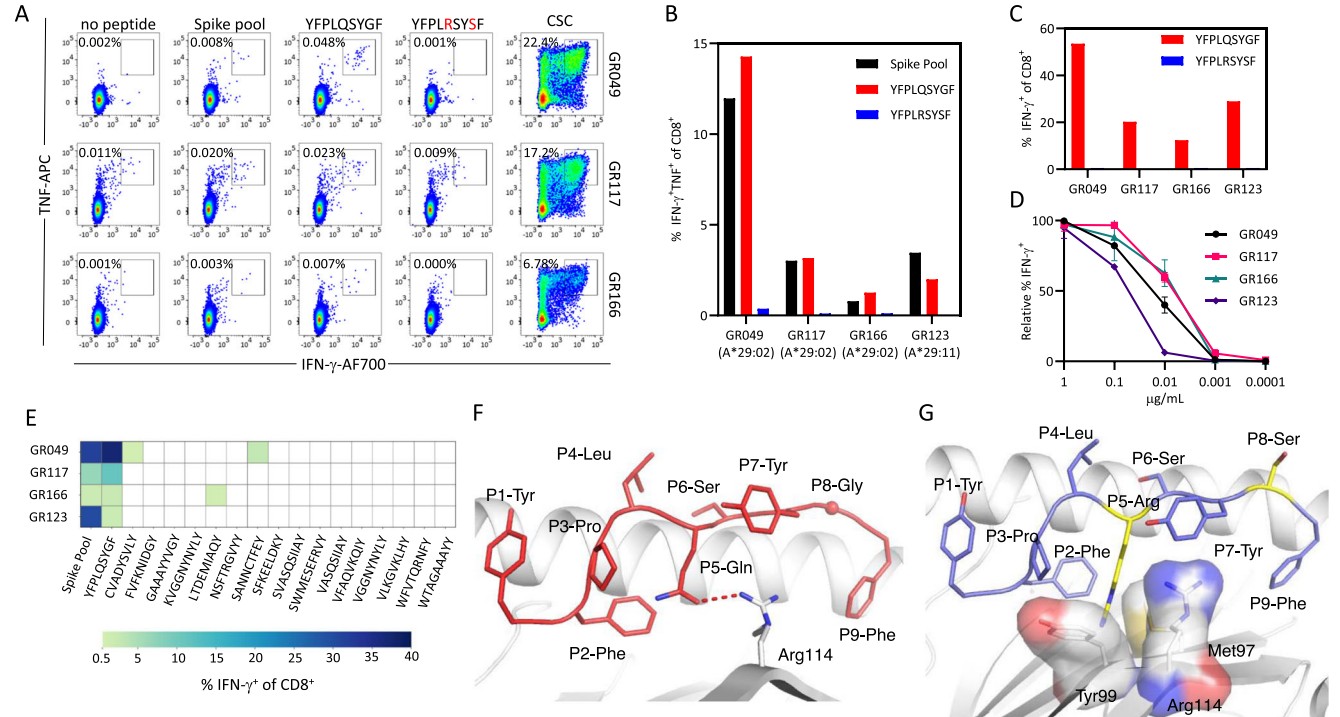

**Fig. 2 | Recognition of the immunodominant HLA-A*29:02-restricted epitope in vaccinated participants. A** Flow cytometry plots displaying the frequency of IFN-γ-producing CD8+ T cells from PBMC of vaccinated HLAA*29:02+ individuals. **B** Frequency of IFN-γ-producing CD8+ T cells following in vitro culture of PBMC from four vaccinated individuals (A*29:02 n = 3; A*29:11 n = 1) with spike overlapping peptide pools. **C** Frequency of IFN-γ-producing CD8+ T cells following in vitro stimulation with either YFPLQSYGF or YFPLRSYSF and recall with cognate peptide. **D** Frequency of IFN-γ-producing CD8+ T cells in response to serial dilutions of YFPLQSYGF. Data represents mean and standard deviation of duplicates. **E** Heatmap of frequency of IFN-γ-producing CD8+ T cells of spike peptide pool,

YFPLQSYGF and published HLA-A29*02 from four vaccinated individuals (A*29:02 n = 3; A*29:11 n = 1). **F** Crystal structure of the HLA-A*29:02-YFPLRSYSF complex with the HLA heavy chain in white cartoon and the peptide in red cartoon and stick. The P8-Gly Cα atom is represented by a sphere, and the hydrogen bond by a red dashed line. **G** Model of the YFP variant peptide (blue cartoon and stick) based on the YFP peptide conformation showing the P5-Arg and P8-Ser substitutions in yellow stick. The HLA Met97, Tyr99, and Arg114 are represented as stick and transparent surface to show the steric clashes with the P5-Arg occurring if the variant adopted the same conformation as the cognate peptide. Source data are provided as a source data file.

T cells from these three participants by stimulating with the spike peptide pools and culturing in the presence of IL-2 for 14 days. In addition, we set up cultures from a fourth individual in our vaccine cohort who is HLA-A*29:11+. Spike-specific CD8+ T cell responses were observed in all four T cell expansions (Fig. 2B, Supplementary Fig. 4B). All four cultures were dominated by T cells specific for YFPLQSYGF, but displayed no recognition of the Omicron variant. In the three HLA-A*29:02+ cultures, the frequency of the YFPLQSYGF-specific CD8+IFN-γ+TNF+ response was comparable to the response against whole spike protein, demonstrating similar immunodominance to that seen in convalescent individuals. To generate variant-specific T cells, we stimulated PBMC with either YFPLRSYSF or the Omicron variant and assessed functional avidity after 14 days in culture. While the Omicron epitope failed to induce the expansion of T cells in these donors, YFPLRSYSF-specific T cells were detected in cultures from all four vaccine recipients (Fig. 2C, D). To confirm the immunodominance of the YFPLQSYGF response in HLA-A*29 individuals, we compared the response to 16 other HLA-A*29:02-restricted epitopes identified from the immune epitope database and prediction resource (IEDB) (Supplementary table 3). While YFPLQSYGF T cell responses were detected in all four volunteers assessed, we could only detect low-frequency subdominant responses to SANNCTFEY (2.83%) and CVADYSVLY (0.66%) in GR049, and LTDMIAQY (0.84%) in GR166, confirming the immunodominance of the YFPLQSYGF in HLA-A29*02-positive individuals (Fig. 2E).

Given the differences in the T cell response between YFPLQSYGF and its variant, we wanted to understand the impact of the Omicron mutation on peptide presentation by HLA-A*29:02. YFPLQSYGF

peptide was refolded with HLA-A*29:02 and then crystallized to understand the basis of peptide presentation (Supplementary Table 4). The HLA-A*29:02 molecule adopted the canonical fold of HLA molecules with YFPLQSYGF[13] (Supplementary Fig. 5A) and in line with the thermal stability observed for the YFP-HLA-A*29:02 complex (Supplementary Table 5).The primary anchors of the YFPLQSYGF peptide at P2 and P9 are both phenylalanine residues that are favored for HLA-A*29:02-peptide binding motif from netMHCpan4.1 website (Fig. 2F). The motif extracted from the netMHCpan4.1 website reports the naturally presented peptide are those which are derived from eluted mass spectrometry data (Supplementary Fig. 6A) however, MHC binder peptides are derived from the binding affinity data from in vitro expansion (Supplementary Fig. 6B). The two residues of the peptide that are mutated in Omicron are at positions 5 and 8. While the substitution at P8 from a glycine to a serine is unlikely to impact peptide presentation and conformation (Fig. 2G), the mutation at position 5 can have an impact on the conformation of the peptide. The P5-Gln is buried in the cleft of HLA-A*29:02, and modeling of a P5-Arg shows a steric clash of the large and charged side chain (Fig. 2G) with residues in the binding cleft due to the presence of large amino acid residues Met97, Tyr99 and Arg114 that occupy the β-sheet floor of HLA-A*29:02. The P5 mutation in Omicron is likely to destabilize the peptide human leukocyte antigen (pHLA) complex and could also weaken its binding to HLA-A*29:02.

To confirm this hypothesis, we refolded the YFP mutant derived from Omicron with the HLA-A*29:02 molecule and performed a stability assay. The result shows a dramatic decrease of the pHLA complex stability by 14 °C due to the mutations in the Omicron-derived peptide

(Supplementary Table 5). This confirmed that the binding of the peptide to HLA-A*29:02 is reduced due to the mutations occurring in the Omicron variant. It is likely that the mutation of the secondary anchor residue P5-Gln is responsible for the decreased Tm value observed, as the P8-Gly is solvent exposed and would not stabilize the peptide but could impact directly on T cell binding. To confirm this hypothesis we have produced single mutant of the YFP peptide at positions 5 and 8 to test the impact of each mutation. Each mutated peptides were able to be refolded with the HLA-A*29:02 molecule, and their stability was determined. The single mutant YFP-P8S in complex with HLA-A*29:02 exhibited a stability close to the one observed for the YFP peptide, decreased by 4 °C (Supplementary Table 5 and Supplementary Fig. 7). Interestingly, both the single mutant YFP-P5R and the omicron variant of YFP both exhibit a similar Tm of ~46–47 °C which is 15 °C lower than the stability of HLA-A*29:02-YFP complex. This shows that indeed the mutation of the P5Q by P5R is destabilizing the pHLA complex.

To investigate the impact of amino acid changes on another spike-encoded epitope, we assessed T cell responses to the HLA-B*07:02-restricted epitope SPRRARSVA[14] in five HLA-B*07:02-positive vaccine recipients (Supplementary Table 2). This epitope contains mutations detectable in multiple variants that alter the HLA-B*07:02 anchor residue at P2 (Supplementary Table 6). While these changes ablated the ability of T cells from vaccinated individuals to recognize the variant peptides, other spike-specific T cell responses were detected, including those against the conserved HLA-A*02:01–restricted epitope YLQPRTFLL[12] (Supplementary Fig. 4C), demonstrating the complex nature of T cell immunodominance in HLA-distinct individuals.

Despite the potentially large number of antigenic targets in the spike protein[12,14,15], our observations suggest that the immunodominance profile in HLA-A29-positive individuals is associated with a skewed response towards a single dominant epitope. While it remains to be determined if this could impact susceptibility in vaccinated individuals, and the current study was limited by the number of individuals assessed, it was clear that changes in the amino acid sequence ablated T cell activation of this immunodominant response. Although HLA-A29 was present in only 6.8% and 3.8% of our convalescent and vaccinated cohort, respectively, HLA-A29 frequencies have been reported as high as 24% in some populations in Africa (allele-frequencies.net, Supplementary Table 7). However, knowledge on the impact of HLA-types on outcome of SARS-CoV-2 infection is currently limited, and the impact of changes in the omicron variants to risk in HLA-A*29 positive individuals requires investigation in a much larger cohort. Nevertheless, these analyses do highlight the impact genetic variation in newly emerging SARS-CoV-2 variants can have upon recognition by immunodominant T-cell responses.

## Methods
### Study participants
This study was approved by the QIMR Berghofer Medical Research Institute Human Research Ethics Committee and was performed according to the principles of the Declaration of Helsinki. All participants were over the age of 18 and provided informed consent to participate in this study. We recruited two cohorts: COVID-19-convalescent participants: COVID-19-recovered donors were over the age of 18, had been clinically diagnosed by PCR with SARSCoV-2 infection, and had subsequently been released from isolation following resolution of symptomatic infection. A total of 58 participants were recruited in May and June 2020 from the south-east region of Queensland, Australia. The majority of participants were returned overseas travelers. Participants ranged in age from 20 to 75, 24 were male and 34 were female, and were a median of 62 (46 – 124) days post-initial diagnosis. Blood samples were collected from all participants to isolate peripheral blood mononuclear cells (PBMCs) to assess SARS-CoV-2 immunity. Healthy donors over the age of 18, with no known COVID-19 infection or exposure, were recruited. These donors are

referred to as unexposed throughout the manuscript. A total of 9 unexposed donors were recruited, ranging in age from 19 to 56 (average of 33 years old), 5 were male, 4 were female. Informed consent was obtained from all participants. The HLA typing was performed by AlloSeq Tx17 (CareDx Pty Ltd, Fremantle, Australia), or Australian Red Cross Victorian Transplant and Immunogenetics Service (Melbourne, Australia), or PathWest Laboratory Medicine, Fiona Stanley Hospital using AllType NGS high resolution typing on the IonTorrent NGS platform, and these details are provided in Supplementary Table 1 and 2.

### Peripheral blood mononuclear cells
Donor peripheral blood mononuclear cells (PBMCs) were separated from whole blood or buffy coats using density gradient centrifugation. PBMC were used fresh or cryogenically stored until use.

### Expansion of SARS-CoV-2-specific T cells
The strategy used for the expansion of SARS-CoV-2-specific T cells has been described[11]. PBMC were stimulated and incubated with 10 µM SARS-CoV-2 overlapping peptide pools or individual peptides expanded for 14 days in RPMI-1640 supplemented with penicillin/streptomycin (Life Technologies) and 10% heat-inactivated fetal bovine serum (Life Technologies). Cultures were supplemented with 10 IU inteleuki-2 every 2–3 days. For the expansion of T cells from convalescent partcipants, pools of 15-mer peptides with an 11-amino-acid overlap were supplied by JPT technologies and are listed in (Supplementary Table 8). SARS-CoV-2-specific peptide matrix pools were also supplied by JPT Technologies. The overlapping peptide pools used to expand T cells from vaccinated participants were supplied by Mimotopes. Individual peptide epitopes used to stimulate T cells were provided by GenScript.

### Intracellular cytokine assay
Cultured T cells ($5 \times 10^5$ per test) or PBMC ($2 \times 10^6$ per test) were stimulated separately with the SARS-CoV-2 overlapping peptide pools (1 µg/mL of each peptide), individual defined epitopes (1 µg/mL) or with a cytokine stimulation cocktail (eBioscience), and incubated for 4 h (T cells) or 6 h (PBMC) at 37 °C in the presence of GolgiPlug (Brefeldin-A), GolgiStop (Monensin) and anti-CD107a-FITC (BD Biosciences). Following stimulation, cells were washed and stained with anti-CD8-PerCP-Cy5.5 (eBioscience), anti-CD4-Pacific Blue (BD Biosciences) and live/dead fixable near-IR dead cell stain (Life Technologies) for 30 min at 4 °C before being fixed and permeabilized with Fixation/Permeabilization solution (BD Biosciences). After 20 min of fixation, cells were washed in BD Perm/Wash buffer (BD Biosciences) and stained with anti-IFN-γ-Alexa Fluor 700, anti-IL-2-PE and anti-TNF-APC (all from BD Biosciences) for a further 30 min at 4 °C. Finally, cells were washed again and acquired using a BD LSRFortessa with FACSDiva software. Post-acquisition analysis was performed using FlowJo software (TreeStar). Cytokine levels detected in the no-peptide control condition were subtracted from the corresponding test conditions to account for non-specific, spontaneous cytokine production.

### Functional avidity assay
T cells ($5 \times 10^5$ per test) were stimulated in duplicate with 10-fold serial dilutions of each peptide epitope, starting at 1 µg/mL, then incubated for 4 h in the presence of GolgiPlug. Cells were then washed and stained with anti-CD8-PerCP-Cy5.5, anti-CD4-FITC (BD Biosciences) and live/dead fixable near-IR dead cell stain for 30 min at 4 °C before being fixed and permeabilized with Fixation/Permeabilization solution for 20 min. Cells were washed with BD Perm/Wash and stained for 30 min at 4 °C with anti-PE-IFN-γ. (BD Biosciences). Cells were washed and acquired using a BD LSRFortessa with FACSDiva software. Post-acquisition analysis was performed using FlowJo software (TreeStar). Cytokine levels detected in the no-peptide control condition were

subtracted from the corresponding test conditions to account for non-specific, spontaneous cytokine production.

## HLA Restriction assay

T cells ($5 \times 10^5$ per test) were stimulated with HLA-A*29:02$^+$ and HLA-A*29:02$^-$ PHA blasts pulsed with 0.1 µg/mL of peptide, then incubated for 4 h in the presence of GolgiPlug. Cells were then washed and stained with anti-CD8-PerCP-Cy5.5, anti-CD4-PE-Cy7 (BD Biosciences) and live/dead fixable near-IR dead cell stain for 30 min at 4 °C before being fixed and permeabilized with Fixation/Permeabilization solution for 20 min. Cells were washed with BD Perm/Wash and stained for 30 min at 4 °C with anti-PE-IFN-γ. (BD Biosciences). Cells were washed and acquired using a BD LSRFortessa with FACSDiva software. Post-acquisition analysis was performed using FlowJo software (TreeStar).

## Protein expression, refold and purification of pHLA complexes

pET-30a$^+$ DNA plasmids encoding HLA-A*29:02 α-chain and human β−2-microglobulin were transformed separately into BL21 E. coli and expressed. Inclusion bodies containing the individually expressed recombinant proteins were extracted, purified and quantified as per previously described (31). Soluble pHLA-A*29:02 complexes were produced by refolding 30 mg of HLA-A*29:02 α-chain with 10 mg of β−2-microglobulin and 5 mg of peptide (Genscript) into a buffer of 3 M Urea, 0.5 M L-Arginine, 0.1 M Tris-HCl pH 8.0, 2.5 mM Ethylenediaminetetraacetic acid (EDTA) pH 8.0, 5 mM glutathione (reduced), 1.25 mM glutathione (oxidized), 0.2 mM Phenylmethylsulfonyl fluoride (PMSF). The refold mixture was dialyzed at 4 °C over 36 h in 10 mM Tris-HCl pH 8.0 and soluble pHLA complexes were purified via anion exchange chromatography. First using a Diethylaminoethyl cellulose resin, then a HiTrapQ column (GE Healthcare).

## Differential scanning fluorimetry

Thermal stability assay was performed by Differential Scanning Fluorimetry carried out in ViiA 7 real-time PCR system (Thermofisher), where pHLA-A*29:02 complexes were heated from 25 to 95 °C at a rate of 1 °C/min in 0.5 °C steps. The excitation and emission channels were set to the TAMRA reporter (x3m3 filter) with excitation of ~550 nm and detection at ~587 nm. The experiment was performed at two concentrations of pHLA (5 µM and 10 µM) in duplicates. Each sample was dialyzed in 10 mM Tris-HCl pH 8.0, 150 mM NaCl and contained a final concentration of 10X SYPRO Orange Dye. Fluorescence intensity data was normalized and plotted using GraphPad Prism 9 (version 9.3). The Tm value for a pHLA is equal to its temperature when 50% of maximum fluorescence intensity is reached, which approximately equals to 50% of unfolded protein and summarized in Supplementary Table 4.

## Crystallization and structure determination

Crystallization of HLA-A*29:02-peptide complex was grown via hanging-drop, vapor diffusion at 20 °C. The protein: reservoir drop ratio was 1:1, at a concentration of 3 mg/mL in 10 mM Tris-HCl pH 8, 150 mM NaCl; and crystals were grown in 1.9 M Ammonium sulfate, 20 mM Magnesium Chordie, 0.1 M Bis-Tris propane, 2% Ethylene glycol, 2% 2-methyl-2,4-pentanediol Protein crystals were soaked in a cryoprotectant solution containing mother liquor solution with 20% (v/v) Ethylene glycol and then flash-frozen in liquid nitrogen. The data were collected on the MX2 beamline at the Australian Synchrotron, part of ANSTO, Australia[16]. The data were processed using XDS[17] and the structures were determined by molecular replacement using the PHASER program[18] from the CCP4 suite[19] with a model of HLA-A*24:02 without the peptide (derived from PDB ID: 7JYV[20]). Manual model building was conducted using COOT[21] followed by refinement with PHENIX, followed by BUSTER[22]. The final models have been validated and deposited using the wwPDB OneDep System and the final refinement statistics, PDB codes are summarized in Supplementary Table 3. All molecular graphics representations were created using PyMOL.

## Synthesis and purification of YFP-P5R and YFP-P8S single mutant peptides

Peptides YFP-P5R: YFPLRSYGF and YFP-P8S: YFPLQSYSF were synthesized using a standard fluoren-9-ylmethoxycarbonyl (Fmoc) automated solid-phase peptide synthesis method with Wang resin (100 – 200 mesh, 1.24 mmol/g) utilizing microwave (MW)-assisted heating on a Biotage® Initiator+ Alstra peptide synthesizer. Prior to use, the Wang resin was swollen in DMF for 2 h. The Fmoc-amino acids and HCTU/HOBt/DIPEA (4 eq.) dissolved in DMF were then added sequentially to the resin and the reactions were carried out with MW heating at 60 °C for 20 min. The Fmoc protecting groups were removed using piperidine (20% in DMF) at RT for 20 min. Peptides were cleaved from the resin using a solution of 95% trifluoroacetic acid (TFA), 5% triisopropylsilane (TIPS) for 3 h. The acid was evaporated, and the crude peptide was purified using RP-HPLC using a Shimadzu HPLC fitted with two Shimadzu LC-20AD pumps, a SIL-20AHT autosampler, a SPD-M20A photodiode array detector and a FRC-10A fraction collector and an Onyx Monolithic C18, 100 × 10 mm semi-preparative column with a gradient of 0.1% TFA in water (solvent A) and acetonitrile (solvent B) over 30 min. Purified peptides were lyophilized and stored at −20 °C. Purity was confirmed at >95% in each case by analytical HPLC and structures were confirmed using high-resolution electrospray ionization mass spectrometry.

## List of reagents

The source, catalog number and if appropriate, dilution of reagents including antibodies is provided in supplementary Table 9.

## Reporting summary

Further information on research design is available in the Nature Research Reporting Summary linked to this article.

## Data availability

All data that support the findings of this study are available in the article and its Supplementary files or upon request from the corresponding author. Source data are provided with this paper. The final crystal structure models for the peptide-HLA-A*29:02 complexes have been deposited to the protein Data Bank (PDB) under the following accession code: 7TLT for HLA-A29-YFPLQSYGF. The SARS-CoV-2 omicron lineage data was retrieved from GISAID database (https://gisaid.org/) under accession ID EPI_ISL_9049930 (BA.1), EPI_ISL_13059703 (BA.2), EPI_ISL_12650713 (BA.3), EPI_ISL_14858758 (BA.4), EPI_ISL_12780920 (BA.5) respectively. Source data are provided with this paper.

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

## Acknowledgements

This work was supported by generous donations to the QIMR Berghofer COVID-19 appeal and the Medical Research Future Fund (MRFF, APP2005654). SS is supported by Australian Government Research Training Program Scholarship and QIMR Berghofer Top-Up Scholarship award. SG is an NHMRC SRF-A Fellow (1159272). KRS is supported by an NHMRC Investigator Grant (2007919). We would like to thank all of the participants who generously donated their blood for this study. This research was undertaken in part using the MX2 beamline at the Australian Synchrotron, part of ANSTO, and made use of the Australian Cancer Research Foundation (ACRF) detector.

## Author contributions

S.S., K.E.L., G.R.A., M.A.N., K.S., S.G., R.K., and C.Sm. contributed to the design of the study. S.S., K.E.L., A.P., J.R., L.D.M., C.Sz. P.C., L.L.T., M.J.D.O., P.J.B. and C.Sm. performed experiments. M.A.N. and S.R. managed the recruitment of participants. All authors contributed to the drafting of the manuscript.

## Competing interests

The authors declare no competing interests.
