## [Peer Review File · Nature Communications]

Ablation of CD8+ T cell recognition of an
immunodominant epitope in SARS-CoV-2 Omicron variants
BA.1, BA.2 and BA.3REVIEWER COMMENTS

Reviewer #1 (Remarks to the Author):

Regarding Immunologic Significance, both the wild type and omicron peptide are predicted to bind A*29:02 well by algorithms such as those available at IEDB. Somewhat surprisingly, the authors find in a crystal structure that the P5 Gln side chain in the wild type peptide participates in HLA binding, and predict that the Arg substitution would lead to steric hindrance. Thus, the authors suggest, but do not prove, that the generally accepted model for peptide binding to A*29:02 is incomplete. As noted below, the authors do not formally study binding of the omicron peptide to A*29:02.

- Medical Significance is not necessarily proven. The authors show responses to this peptide in 3 of 4 A*29:02 persons, but population prevalence is different from immunodominance, and this is a small number of individuals. The paper states that in convalescent or vaccinated A*29:02 persons exposed to wild-type S, the particular epitope studied is “immunodominant”. We are not shown evidence that response to this peptide, within person, in vivo, are numerically abundant compared to responses to other peptides in S. The authors use epitope mapping after a boost with long peptides (15 mers), which may not really reflect the in vivo situation. It is well known for example that some CD8 cells will respond to long peptides that are longer than the minimal fully active 8-10mers, while other CD8 responses are abrogated if the peptide is even one AA too long. I’d like ex vivo S protein epitope mapping in A*29:02 persons to make the argument that the sole and/or numerically dominant CD8 responses in such persons are to this epitope. If this were true Medical Significance would be bolstered.

- The millions of persons recently infected with Omicron include A*29:02 (+) persons with or without prior vaccination. If the hypothesis that the omicron peptide does not bind A*29:02 well is correct, we’d predict these people would not have CD8 responses to the Omicron variant peptide. We might also predict that total CD8 T cell responses to the Omicron peptide set covering the full length of Omicron S would be low in A*29:02 persons. Medical Significance would be added if response to the variant peptide and optimally to full-length Omicron was studied in A*29:02 Omicron-convalescent persons. In this regard, several recent publications conducted ex vivo or in silico do not find significant alterations in the overall T cell response to wild type vs. omicron S protein, for example PMID 35139340, 35113647, 35102312, 35102311

Nature family publications typically include some sort of expanded or star methods. The methods as provided are not in enough detail to allow replication but they are reasonable to allow evaluation.

Major:

Physical binding studies of the omicron variant to A*29:02 would reinforce predictions based on the complex of A*29:02 and the wild-type peptide that and P5 Q to R change reduced peptide binding to HLA. This should be repeated with peptides with single mutations at P5nad P8 as well as the double. These assays can be done by competition for example and are widely available.

Similarly, the CD8 T cell functional assays in Figs 1 and 2 should investigate the structural basis somewhat more in detail by testing the single AA mutant peptides.

The material in lines 126-134 is unrelated to the structural biology focus of the rest of the paper and does not really speak to immunodominance. For example the A*02:01 YLQ epitope (S 269-277) is probably the best studied SARS CoV 2 CD8 epitope at this point. We are not really given evidence that the A*29:02 epitope studied in this report is so immunodominant that in A*02:01/A*29:02 heterozygotes that the A*29:02 epitope “wins”. Just the fact that responses to an A2 and B7 restricted epitope can co-exist in A2/B7 vaccinees does not mean that in A*29:02 persons, other responses to Spike are not seen. Along these lines, it is puzzling that the vast majority of the non-A29 convalescent

people in Fig 1A seem to have no CD8 T cell response to either S1 or S2 using peptide pools representing wild-tuype

Minor:

Fig 1 seems to all use culture expanded cell lines made by incubating PBMC with the wild type peptide. While this is briefly mentioned in Results this should also be called out in the Figure legend.

Line 124-125 sentence is missing a word and thus the meaning is unclear.

Line 79 use generic names that describe the vaccine, not proprietary names. The vaccines used were a rep. incomp. chimp adeno or an mRNA. This is important as some vaccine formats e.g. protein or killed virus used worldwide are expected to not elicit CD8 T cells.

Line 259 use chemical names not brand names for Golgi-X.

Cohort please clarify the approximate time frame the convalescent donors were infected with SARS-CoV-2: was it before the recent Omicron wave?

Possibly the Omicron peptide is not well recognized because the Omicron sequence is identical to, or very close to a human peptide to which CD8 responses should be low. A scan of the predicted human proteome would examine this hypothesis.

Reviewer #2 (Remarks to the Author):

In the manuscript Ablation of CD8+ T-cell recognition of an immunodominant epitope in SARS-CoV-2 Omicron, Swaminathan et al describe an immunodominant epitope for HLA-A29 derived from SARS-CoV-2 spike protein, structurally characterize it, and demonstrate a loss of immunogenicity (or at least cross-reactivity with the original immune response) associated with a Gln to Arg mutation found in the SARS-CoV-2 omicron variant. This work is largely well-conducted, but there is not enough evidence in the data as presented to be sure of the hypothesis offered here.

- The authors convincingly demonstrate that YFPLQSYGF is an immunodominant epitope that binds to HLA-A29*02 (including a structure), and that a Gln to Arg polymorphism is enough to prevent cross-reactivity from convalescent or vaccinated patients to this epitope from the omicron variant. However, to my knowledge the authors do not provide any direct evidence that the loss of immunogenicity arises from loss of peptide binding to MHC beyond some basic modeling. It is still quite plausible that the loss arises from some other factor (such as binding but change in backbone configuration), especially given the NetMHCpan prediction for YFPLQSYGF and YFPLRSYGF are very similar, and there are multiple examples of peptides with a P5 Arg for HLA-A29*02 in the IEDB. Since this is so central to the paper, some orthogonal means of validation (such as a thermal melt assay as done for the original peptide in Supplemental Table 4) should be conducted.

- While it is true that loss of immunodominant epitopes can affect disease response, I find the framing of this issue overstated in this case: essentially every HLA-A29*02 individual responds to multiple antigens, and mitigating these types of escape mutations is precisely the role of HLA diversity even within an individual. Further, to my knowledge there has not been any significant HLA associations for clinical outcomes for SARS-CoV-2. I'd therefore encourage the authors to better contextualize their motivation and take-away sections (lines 45-48, 135-146)

- As written, the inclusion of the HLA-B7 data in Figure 2G is somewhat confusing as it is not linked to the clear focus on HLA-A29 in the rest of the manuscript beyond it being a lost epitope likely due to a relatively straightforward P2 anchor mutation. This data should either be moved to the supplement, removed, or the introduction etc should be expanded to represent a broader look at MHC antigen

escape.

- There are multiple places where the statistics are not sufficiently described to understand the data and/or to comply with Nature guidelines: either places where there are no error bars (such as 2G), or where the error bars are not defined (1H). The thermal midpoint measurement also seems to derive as SEM from 2 measurements, which I do not believe is correct.
- The authors include one HLA-A*29:11 patient, but with no description in the text. I presume this is quite closely related to HLA-A*29:02, but this should be explicitly said.
- Could the authors further clarify the differences between the data derived for Supplementary Figure 6A and 6B?
- In supplementary table 3, there is a symbol missing for I/sigI

Reviewer #3 (Remarks to the Author):

Swaminathan et al identify an immunodominant T-cell epitope in HLA-A29 positive individuals. This CD8 T-cells epitope varies at two positions in the Omicron variant of CoV-2, and these changes at P5 and P8 ablate recognition by the epitope-specific T-cells. The authors went on to show that this epitope is immunodominant in HLA-A29 positive individuals and that it binds to HLA-A29. Overall, the study is well done, the conclusions are supported by the data, and the manuscript is well written.

My main concern is the small number of HLA-A29 positive individuals, which are also B44 positive, and the claim that these individuals do not respond to any other CD8 T-cell epitopes in the S1 subunit of the Spike protein. The small number of HLA29 individuals is addressed using a second cohort of vaccinated individuals with a slightly more diverse HLA-B alleles. Three or four of these individuals are also A29 positive individuals and again the epitope is immunodominant in their assays. However, the claim that these individuals do not respond to other epitopes in S1 is inferred from the fact that the % IFN-gamma positive cells after S1 peptide pool stimulation are identical between the epitope and the S1 pool. Is this because the S1 subunit is a relatively small (~220 amino-acids)? What is the role of B44 in 5/7 HLA-A29 positive individuals? Is this a dominant allele whose epitope is outside of the S1 subunit? To suggest that the T-cells from infected or vaccinated HLA-A29 individuals cannot recognize the Omicron variant, the authors should stimulate the PBMC with S1 + S2 peptide pools and identify what portion of the Spike protein response is towards this single epitope.

This epitope was previously found in A24 positive individuals? A29 is part of the A01/A24 superfamily. Is this epitope recognized or immunodominant in these individuals in this cohort?

Perhaps not part of this study, it would have been very interesting to identify the TCR repertoire of epitope specific T-cells. Follow up studies would have been able to identify cross-reactive T-cells or omicron-specific T-cells and the authors could have started looking at MHC-peptide-TCR complexes and their rules of engagement.

We thank all of the reviewers for their helpful comments. A point by point response to each reviewer comment is provided below.

Reviewer#1:

1. Regarding Immunologic Significance, both the wild type and omicron peptide are predicted to bind A*29:02 well by algorithms such as those available at IEDB. Somewhat surprisingly, the authors find in a crystal structure that the P5 Gln side chain in the wild type peptide participates in HLA binding, and predict that the Arg substitution would lead to steric hindrance. Thus, the authors suggest, but do not prove, that the generally accepted model for peptide binding to A*29:02 is incomplete. As noted below, the authors do not formally study binding of the omicron peptide to A*29:02.

Response: We would like to thank the reviewer for these comments, however we would like to point out that we are not necessarily trying to suggest that the accepted model for peptide binding is A*29:02 is incorrect. Rather we are stating that in the peptide we have identified, the P5-Gln is acting as a secondary anchor residue which is a very common finding in HLA-I peptide binding (DOI: 10.1042/BST20210410). Furthermore, there are no previous structures defined for HLA-A29, most data on HLA-A29 restricted epitopes is limited to predictions and there is a lack of peptide-elution and identification by mass spectrometry directly from antigen presenting cells. As a consequence there is very limited data on this HLA-A29 molecule and it is not clear if similar secondary anchors are present in other peptides. The presence of a secondary anchor is not always a requirement for binding and could be specific to certain peptides, like the one studied here. We have now added the data from the stability assay for the omicron peptide, showing the dramatic decrease of pHLA complex stability due to the mutation at P5, further evidence that it is functioning as a secondary anchor. With regard to the remaining structural analysis presented here, the data does in fact agree with the current limited data available for HLA-A29. We show that the binding motif contains a P2-F and P-omega-F residue as preferred as primary anchors, as expected for HLA-A29 epitopes.

2. Medical Significance is not necessarily proven. The authors show responses to this peptide in 3 of 4 A*29:02 persons, but population prevalence is different from immunodominance, and this is a small number of individuals.

Response: We do agree with the reviewer that the small number of HLA-A29 volunteers in our study limits the potential to assess medical significance. Our analysis was limited by the low prevalence of HLA-A29 in Australia. However, we would like to point out that at the stage of preparation of this data, most analysis had suggested that T cell immunity was not likely to be affected by amino acid substitutions present in omicron. Our observations provide evidence to counter this generalisation and demonstrate that there can be a significant impact in the context of particular HLA alleles that drive highly biased immunodominant responses.

3. The paper states that in convalescent or vaccinated A*29:02 persons exposed to wild-type S, the particular epitope studied is "immunodominant". We are not shown evidence that response to this peptide, within person, in vivo, are numerically abundant compared to responses to other peptides in S. The authors use epitope mapping after a boost with long peptides (15 mers), which may not really reflect the in vivo situation. It is well known for example that some CD8 cells will respond to long peptides that are longer than the minimal fully active 8-10mers, while other CD8 responses are abrogated if the peptide is even one AA too long. I'd like ex vivo S protein epitope mapping in A*29:02 persons to make the argument that the sole and/or numerically dominant CD8 responses in such persons are to this epitope. If this were true Medical Significance would be bolstered.

Response: We politely disagree with the reviewer on this point. The use of overlapping 15-mer peptides is a very standard approach that has proven to be very effective at mapping immunodominant epitopes in human viral pathogens (including EBV, CMV, BKV). It is a particularly important approach used to find epitopes that are not identified using standard binding algorithms, such as the YFPLQSYGF defined in this study that was not previously shown to be restricted to A*29. We would also like to point out that in our extensive experience we have not seen evidence of 15-mer peptides failing to recall immunodominant CD8+ T cell responses, and these

longer peptides are likely processed into the correct length of 8-9 amino acids for presentation. To further demonstrate the immunodominance of the YFPLQSYGF response in HLA-A*29 individuals, we have now compared the response to 16 other previously reported HLA-A*29 epitopes from the spike protein, demonstrating a low prevalence of T cell responses to these other peptides.

4. The millions of persons recently infected with Omicron include A*29:02 (+) persons with or without prior vaccination. If the hypothesis that the omicron peptide does not bind A*29:02 well is correct, we'd predict these people would not have CD8 responses to the Omicron variant peptide. We might also predict that total CD8 T cell responses to the Omicron peptide set covering the full length of Omicron S would be low in A*29:02 persons. Medical Significance would be added if response to the variant peptide and optimally to full-length Omicron was studied in A*29:02 Omicron-convalescent persons. In this regard, several recent publications conducted ex vivo or in silico do not find significant alterations in the overall T cell response to wild type vs. omicron S protein, for example PMID 35139340, 35113647, 35102312, 35102311

Response: We agree that studying the responses following infection of HLA-A*29 individuals with Omicron would be of great value. However, in order to do this we would likely need to look towards assessing these responses in populations outside of Australia, where the prevalence of HLA-A29 is more common. We currently do not have ethical approval for this work. At the time of recruitment for the current study (Dec 2021), infection rates in Australia were very low and we had few individuals exposed to omicron. As part of follow-up work we hope to expand our cohort to assess responses to infection post-vaccination, depending on success of grant funding. However, we would also like to point out that our new data on the stability of the omicron variant of the YFP peptide demonstrates that the mutations within the HLA A29-restricted epitope sequence have a dramatic impact on peptide stability and the variant peptide itself is therefore unlikely to prime a strong response following infection.

The recent publications which the reviewer mentioned above (PMID 35139340, 35113647, 35102312, 35102311) do not report on HLA-A29 peptide.

- (1) PMID 35139340 – This manuscript does not include HLA-A29 peptides. This study is based on the USA cohort where HLA-A29 is low.
- (2) PMID 35113647 – No HLA typing mentioned and the peptides are screened using overlapping peptides.
- (3) PMID 35102312 - No HLA typing mentioned and the peptides are screened using overlapping peptides.
- (4) PMID 35102311 - No HLA typing mentioned and the peptides are screened using overlapping peptides.

Nature family publications typically include some sort of expanded or star methods. The methods as provided are not in enough detail to allow replication but they are reasonable to allow evaluation.

Thank you for this feedback, we will have the star methods prepared if the manuscript gets accepted.

Major:

5. Physical binding studies of the omicron variant to A*29:02 would reinforce predictions based on the complex of A*29:02 and the wild-type peptide that and P5 Q to R change reduced peptide binding to HLA. This should be repeated with peptides with single mutations at P5nad P8 as well as the double. These assays can be done by competition for example and are widely available. Similarly, the CD8 T cell functional assays in Figs 1 and 2 should investigate the structural basis somewhat more in detail by testing the single AA mutant peptides.

Response: We agree with the reviewer on this and have now generated peptides that contain single amino acid substitutions at position 5 and 8 in order to assess the potential impact of each change on T cell recognition. We initially performed a thermal stability assay with these single amino substitutions and compared this to the stability of the wild type and omicron variant. YFPLQSYSF. While the substitution at P8 did not affect stability, the single mutant YFPLRSYGF and the YFPLRSYSF omicron variant exhibit a similar melting temperature of ~46-47°C which is 15°C lower than the stability of HLA-A*29:02-YFP complex. This demonstrates that the point mutation P to R at P5 destabilizes the pHLA complex. We also performed T cell

functional avidity assay on mutated peptides. Both the single P5 and P8 substitutions ablated the T cell response, suggesting that P8 is likely a TCR contact residue. These results are included in the Supplementary figure 3C, 3D, 3E.

6. The material in lines 126-134 is unrelated to the structural biology focus of the rest of the paper and does not really speak to immunodominance. For example the A*02:01 YLQ epitope (S 269-277) is probably the best studied SARS CoV 2 CD8 epitope at this point. We are not really given evidence that the A*29:02 epitope studied in this report is so immunodominant that in A*02:01/A*29:02 heterozygotes that the A*29:02 epitope “wins”. Just the fact that responses to an A2 and B7 restricted epitope can co-exist in A2/B7 vaccinees does not mean that in A*29:02 persons, other responses to Spike are not seen. Along these lines, it is puzzling that the vast majority of the non-A29 convalescent people in Fig 1A seem to have no CD8 T cell response to either S1 or S2 using peptide pools representing wild-type

Response: We agree with the reviewer on this. However, we do not have any volunteers who share a response to the YLQ and YFP epitope. However, we would like to point out that our observations demonstrate that in the HLA-A*29:02 individuals, the response to the YFP epitope was comparable to the total spike response. To further demonstrate YFPLQSYGF immunodominance, we identified sixteen spike encoded HLA-A29*02 epitopes reported in IEDB, and assessed the responses to these epitopes in T cells from HLA-A29 vaccinated donors. Compared to the YFPLQSYGF epitope, these other epitopes were rarely detected and at much lower frequencies. These results are included in Figure 2E.

Minor:

7. Fig 1 seems to all use culture expanded cell lines made by incubating PBMC with the wild type peptide. While this is briefly mentioned in Results this should also be called out in the Figure legend.

Response: We thank the reviewer for this comment and have now included this information in the Figure legend.

8. Line 124-125 sentence is missing a word and thus the meaning is unclear.

Response: We have now corrected this error. The P5 mutation in Omicron is likely to destabilize the pHLA complex and could also weaken its binding to HLA-A*29:02.

9. Line 79 use generic names that describe the vaccine, not proprietary names. The vaccines used were a rep. incomp. chimp adeno or an mRNA. This is important as some vaccine formats e.g. protein or killed virus used worldwide are expected to not elicit CD8 T cells.

Response: We have now corrected this error

From a cohort of 104 participants, recruited following vaccination with either adenovirus vector (Vaxzevria) or an mRNA (Cominarty), we identified three participants with HLA-A*29:02

10. Line 259 use chemical names not brand names for Golgi-X.

Response: We have now corrected this error

11. Cohort please clarify the approximate time frame the convalescent donors were infected with SARS-CoV-2: was it before the recent Omicron wave?

Response: The convalescent donors were recruited 46-131 days after initial diagnosis following the first wave of infection in Australia. This was prior to any omicron outbreak in Australia. Vaccinated individuals were recruited 28 days after the 2nd dose of vaccine. This was prior to the omicron outbreak and when exposure was very low in Brisbane, Australia, where donors were recruited from.

12. Possibly the Omicron peptide is not well recognized because the Omicron sequence is identical to, or very close to a human peptide to which CD8 responses should be low. A scan of the predicted human proteome would examine this hypothesis.

Response: We agree that there is always a possibility for sequence overlap with self-antigens, however we believe our data clearly now shows that the reduced binding stability of the omicron variant to HLA-A29 is most likely the cause of the poor recognition.

Reviewer #2

1. In the manuscript Ablation of CD8+ T-cell recognition of an immunodominant epitope in SARS-CoV-2 Omicron, Swaminathan et al describe an immunodominant epitope for HLA-A29 derived from SARS-CoV-2 spike protein, structurally characterize it, and demonstrate a loss of immunogenicity (or at least cross-reactivity with the original immune response) associated with a Gln to Arg mutation found in the SARS-CoV-2 omicron variant. This work is largely well-conducted, but there is not enough evidence in the data as presented to be sure of the hypothesis offered here.

Response: We would like to thank the reviewer for providing the feedback on the manuscript. To confirm this hypothesis, we refolded the YFP mutant derived from Omicron with the HLA-A*29:02 molecule and performed a stability assay. The result shows a dramatic decrease of the pHLA complex stability by 14°C due to the mutations in the Omicron derived peptide (Supplementary Table 5). This confirmed that the binding of the peptide to HLA-A*29:02 is reduced due to the mutations occurring in the Omicron variant. It is likely that the mutation of the secondary anchor residue P5-Gln is responsible for the decreased T_m value observed, as the P8-Gly is solvent exposed and would not play a major role in stabilizing the peptide but could impact directly on T cell binding. To confirm this hypothesis we have produced single mutant of the YFP peptide at positions 5 and 8 to test the impact of each mutation. Each mutated peptides were able to be refolded with the HLA-A*29:02 molecule, and their stability was determined. The single mutant YFP-P8S in complex with HLA-A*29:02 exhibited a stability close to the one observed for the YFP peptide, decreased by 4°C (Supplementary Table 5 and Supplementary Figure 7). Interestingly, both the single mutant YFP-P5R and the omicron variant of YFP both exhibit a similar T_m of ~46-47°C which is 15°C lower than the stability of HLA-A*29:02-YFP complex. This shows that indeed the mutation of the P5Q by P5R is destabilizing the pHLA complex.

2. The authors convincingly demonstrate that YFPLQSYGF is an immunodominant epitope that binds to HLA-A29*02 (including a structure), and that a Gln to Arg polymorphism is enough to prevent cross-reactivity from convalescent or vaccinated patients to this epitope from the omicron variant. However, to my knowledge the authors do not provide any direct evidence that the loss of immunogenicity arises from loss of peptide binding to MHC beyond some basic modeling. It is still quite plausible that the loss arises from some other factor (such as binding but change in backbone configuration), especially given the NetMHCpan prediction for YFPLQSYGF and YFPLRSYGF are very similar, and there are multiple examples of peptides with a P5 Arg for HLA-A29*02 in the IEDB. Since this is so central to the paper, some orthogonal means of validation (such as a thermal melt assay as done for the original peptide in Supplemental Table 4) should be conducted.

Response: We would like to thank the reviewer for the comments. We have now included thermal stability data in the supplementary data to show reduced binding of the variant to A*29:02.

3. While it is true that loss of immunodominant epitopes can affect disease response, I find the framing of this issue overstated in this case: essentially every HLA-A29*02 individual responds to multiple antigens, and mitigating these types of escape mutations is precisely the role of HLA diversity even within an individual. Further, to my knowledge there has not been any significant HLA associations for clinical outcomes for SARS-CoV-2. I'd therefore encourage the authors to better contextualize their motivation and take-away sections (lines 45-48, 135-146)

Response: We agree that HLA diversity offers the opportunity to mitigate mutations that impact specific T cell responses. However, it was clear in our A*29:02 volunteers that the response to other epitopes in vaccinated individuals, including other A*29:02 epitopes were low if detectable at all and that escape from this epitope will impact vaccine induced T cell recognition to omicron. However, we also agree that there has been no definitive associations between particular HLA alleles and risk of disease following vaccination and it is likely that the complexity of the response in most individuals will overcome the impact of these changes. We have now added some additional conclusions to reiterate that the impact of these changes on risk of disease are not yet elucidated.

4. As written, the inclusion of the HLA-B7 data in Figure 2G is somewhat confusing as it is not linked to the clear focus on HLA-A29 in the rest of the manuscript beyond it being a lost epitope likely due to a relatively straightforward P2 anchor mutation. This data should either be moved to the supplement, removed, or the introduction etc should be expanded to represent a broader look at MHC antigen escape.

Response: We have now moved the HLA-B7 data to supplementary figure 4C. The objective of including this data was to demonstrate that changes in omicron can affect other responses, however in the A2 B7 individuals it was clear that other dominant responses were present following vaccination that may restrict the impact of mutation by providing recognition through other immunodominant epitopes.

5. There are multiple places where the statistics are not sufficiently described to understand the data and/or to comply with Nature guidelines: either places where there are no error bars (such as 2G), or where the error bars are not defined (1H). The thermal midpoint measurement also seems to derive as SEM from 2 measurements, which I do not believe is correct

Response: We have provided errors bars when multiple replicates were used. Due to limited availability of sample, measurements were often performed as a single replicate.

6. The authors include one HLA-A*29:11 patient, but with no description in the text. I presume this is quite closely related to HLA-A*29:02, but this should be explicitly said.

Response:

With regard to Figure 2B, the reason for including A*29:11 donor is to understand if the YFPLQSYGF is also presented on a different subtype other than A*29:02 and if we would see a similar impact of the omicron variant. While A*29:11 and other A29 subtypes are rare in Australia, they are present in other regions of the world, particularly Africa, where A29 is more common. We therefore thought it was important to include this as confirmation that the omicron mutation has potential to affect recognition in people with different A29 subtypes.

7. Could the authors further clarify the differences between the data derived for Supplementary Figure 6A and 6B?

Response: The supplementary figure 6A and 6B, suggests that the predicted **peptide binding motif from netMHCpan4.1 website** correlates with our molecular binding of YFPLQSYGF peptide to HLA-A*29:02 molecule. The primary anchors of the YFPLQSYGF peptide at P2 and P9 are both phenylalanine residues that are favoured for HLA-A*29:02 peptide binding motif. The motif have been extracted from the netMHCpan4.1 web site reporting the 6A: naturally presented peptide are those which are derived from eluted mass spectrometry data, whereas 6B: MHC binder peptides are derived from the binding affinity data from in vitro expansion. Therefore the data demonstrates that both approaches enrich for the anchors present in our peptide

8. In supplementary table 3, there is a symbol missing for I/sigI

Response: We have corrected in the supplementary table.

Reviewer #3

1. Swaminathan et al identify an immunodominant T-cell epitope in HLA-A29 positive individuals. This CD8 T-cells epitope varies at two positions in the Omicron variant of CoV-2, and these changes at P5 and P8 ablate recognition by the epitope-specific T-cells. The authors went on to show that this epitope is immunodominant in HLA-A29 positive individuals and that it binds to HLA-A29. Overall, the study is well done, the conclusions are supported by the data, and the manuscript is well written.

Response: We would like to thank the reviewer for their feedback on the manuscript.

2. My main concern is the small number of HLA-A29 positive individuals, which are also B44 positive, and the claim that these individuals do not respond to any other CD8 T-cell epitopes in the S1 subunit if the Spike protein. The small number of HLA29 individuals is addressed using a second cohort of vaccinated individuals with a slightly more diverse HLA-B alleles. Three or four of these individuals are also A29 positive individuals and again the epitope is immunodominant in their assays. However, the claim that these individuals do not respond to other epitopes in S1 is inferred from the fact that the % IFN-gamma positive cells after S1 peptide pool stimulation are identical between the epitope and the S1 pool. Is this because the S1 subunit is a relatively small (~220 amino-acids)? What is the role of B44 in 5/7 HLA-A29 positive individuals? Is this a dominant allele whose epitope is outside of the S1 subunit? To suggest that the T-cells from infected or vaccinated HLA-A29 individuals cannot recognize the Omicron variant, the authors should stimulate the PBMC with S1 + S2 peptide pools and identify what portion of the Spike protein response is towards this single epitope.

Response: All volunteers assessed in the study were stimulated with overlapping peptides that encompass the full length spike protein, not just the S1 subunit. We were not able to detect any significant other reactivities against spike peptides/epitopes from the A*29:02 volunteers other than what is shown in the manuscript. HLA B44 is commonly shared as part of a haplotype with A*29:02. We did not detect any B44 epitope responses in these volunteers.

3. This epitope was previously found in A24 positive individuals? A29 is part of the A01/A24 superfamily. Is this epitope recognized or immunodominant in these individuals in this cohort?

Response: We did not see evidence of responses to this epitope in other HLA types in our cohort.

4. Perhaps not part of this study, it would have been very interesting to identify the TCR repertoire of epitope specific T-cells. Follow up studies would have been able to identify cross-reactive T-cells or omicron-specific T-cells and the authors could have started looking at MHC-peptide-TCR complexes and their rules of engagement.

Response: We agree with the reviewer that studies on TCR usage would be highly valuable. We are currently planning this work for follow-up studies.

Please note: Changes in the manuscript text are boxed in yellow. We have also added some update information on sequence changes in more recent Omicron variants in Figure 1G.

REVIEWERS' COMMENTS

Reviewer #1 (Remarks to the Author):

Changes adequately address critiques.

Reviewer #2 (Remarks to the Author):

I thank the authors for their experiments and clarifications on the text. I think the data provided is sufficient for publication. However, I would still encourage further changes to the text to 'soften' or better contextualize the claim that the described spike mutations mean that BA1/BA2 variants of SARS-CoV-2 may evade immunity in HLA-A29+ individuals.

The data presented here remains necessary but not sufficient for this claim. In a rough increasing level of sufficiency, to make this claim:

"The P5 and P8 mutations of the peptide ablate all/most of the T cell response against the YF9 antigen in individuals who have been exposed to vaccines or pre-Omicron variants."

The paper indeed convincingly shows this with orthogonal approaches.

"The mutations at P5/P8 would prevent a newly derived immune response against this epitope."

This is plausible, and perhaps even likely, given the data shown Supplementary Figure 7 demonstrating a decrease in stability due to P5 Gln \diamond Arg, but is not quite rigorously proven here (this would require a peptide expansion using this epitope in a patient who was exposed to BA1/BA2).

"The loss of an immunodominant, HLA-A29-restricted T cell response impairs overall T cell immunity in HLA-A29+ individuals."

This is also possible, although there is only indirect data supporting this claim presented here, and overall seems conceptually uncertain. This is for a couple of reasons. First, it can be difficult to measure overall magnitude of T cell responses from the expansion and stimulation assays done here, as compared to the relative composition of responses. Something like an ELISPOT assay conducted on unexpanded T cells with a D614G or Delta sequence as compared to BA2 would be a way of doing this, but even there it is quite likely that a range of overall T cell responses would be found due to inter-patient variability. This is especially the case as I do not think that any T cell expansions were done with a BA1/2 equivalent of a spike pool. Second, the HLA heterogeneity of patient populations makes it seem unlikely that loss of a single allele would impair immunity. As a thought experiment: if an HLA-A29/HLA-A2 heterozygote were compared to an HLA-A2 homozygote, wouldn't the ablation of an HLA-A29-restricted response reduce the heterozygous individual to the same epitope availability as the (protected) HLA-A2 homozygote? It is possible that a previous exposure or vaccination could skew the recognition repertoire, but even that is far from saying that T cell function would be impaired in HLA-A29+ individuals overall (including those who had not been exposed).

"The change in HLA-A29-restricted immunity leads to a change in patient outcomes."

To my knowledge, there is simply no data suggesting this either in this paper or anywhere else.

All of this is not to doubt the data presented here, or its worthiness of publication – I support both! But given how important a topic this is, I would ask the authors to further reign in the most striking claims, particularly in the Abstract ("We also failed to detect other immunodominant responses in vaccinated HLA-A*29:02+ individuals, suggesting that the Omicron variant may have an increased capacity to escape immune recognition in this population."), and in the last few lines of the discussion ("While it remains to be determined if this could impact susceptibility in vaccinated individuals, it was clear that changes in the amino acid sequence ablated T cell activation." – this is true, but for the T cell responses that were restricted to this antigen), and ("Demonstrating that ongoing genetic variation may lead to escape from cellular immune control, especially in ethnic groups in which HLA alleles impacted by epitope mutations are dominant.")

Reviewer #3 (Remarks to the Author):

The authors have responded very well to the reviewers critiques and comments. The inclusion of the stability assay and showing that P5 is important for the stability of the complex is a key piece of data needed to support their model and conclusions.

Thank you for adding a sentence to the discussion addressing point 3 of Reviewer #2. However, to further nuance the paper and finding, I think it is also important to acknowledge in the title and/or abstract that this change was only found in BA.1 and BA.2 Omicron and NOT BA.4 and BA.5.

We thank all of the reviewers for their helpful comments. A point by point response to each reviewer comment is provided below.

REVIEWERS' COMMENTS

Reviewer #1 (Remarks to the Author):

Changes adequately address critiques.

Response: We thank the reviewer for their feedback.

Reviewer #2 (Remarks to the Author):

I thank the authors for their experiments and clarifications on the text. I think the data provided is sufficient for publication. However, I would still encourage further changes to the text to ???soften??? or better contextualize the claim that the described spike mutations mean that BA1/BA2 variants of SARS-CoV-2 may evade immunity in HLA-A29+ individuals.

The data presented here remains necessary but not sufficient for this claim. In a rough increasing level of sufficiency, to make this claim:

???The P5 and P8 mutations of the peptide ablate all/most of the T cell response against the YF9 antigen in individuals who have been exposed to vaccines or pre-Omicron variants.???

The paper indeed convincingly shows this with orthogonal approaches.

???The mutations at P5/P8 would prevent a newly derived immune response against this epitope.???

This is plausible, and perhaps even likely, given the data shown Supplementary Figure 7 demonstrating a decrease in stability due to P5 Gln ??? Arg, but is not quite rigorously proven here (this would require a peptide expansion using this epitope in a patient who was exposed to BA1/BA2).

???The loss of an immunodominant, HLA-A29-restricted T cell response impairs overall T cell immunity in HLA-A29+ individuals.???

This is also possible, although there is only indirect data supporting this claim presented here, and overall seems conceptually uncertain. This is for a couple of reasons. First, it can be difficult to measure overall magnitude of T cell responses from the expansion and stimulation assays done here, as compared to the relative composition of responses. Something like an ELISPOT assay conducted on unexpanded T cells with a D614G or Delta sequence as compared to BA2 would be a way of doing this, but even there it is quite likely that a range of overall T cell responses would be found due to inter-patient variability. This is especially the case as I do not think that any T cell expansions were done with a BA1/2 equivalent of a spike pool. Second, the HLA heterogeneity of patient populations makes it seem unlikely that loss of a single allele would impair immunity. As a thought experiment: if an HLA-A29/HLA-A2 heterozygote were compared to an HLA-A2 homozygote, wouldn't the ablation of an HLA-A29-restricted response reduce the heterozygous individual to the same epitope availability as the (protected) HLA-A2 homozygote? It is possible that a

previous exposure or vaccination could skew the recognition repertoire, but even that is far from saying that T cell function would be impaired in HLA-A29+ individuals overall (including those who had not been exposed).

???The change in HLA-A29-restricted immunity leads to a change in patient outcomes.???

To my knowledge, there is simply no data suggesting this either in this paper or anywhere else.

All of this is not to doubt the data presented here, or its worthiness of publication ??? I support both! But given how important a topic this is, I would ask the authors to further reign in the most striking claims, particularly in the Abstract (???We also failed to detect other immunodominant responses in vaccinated HLA-A*29:02+ individuals, suggesting that the Omicron variant may have an increased capacity to escape immune recognition in this population.???), and in the last few lines of the discussion (???While it remains to be determined if this could impact susceptibility in vaccinated individuals, it was clear that changes in the amino acid sequence ablated T cell activation.??? ??? this is true, but for the T cell responses that were restricted to this antigen), and (???Demonstrating that ongoing genetic variation may lead to escape from cellular immune control, especially in ethnic groups in which HLA alleles impacted by epitope mutations are dominant.???)

Response: We thank the reviewer for their helpful feedback. We have made changes in the abstract and discussion section to the manuscript in order to reduce any implications that mutation in the A29 epitope is having any impact on patient outcome, and have instead focused on the clear fact that the changes were ablating recognition by T cells raised the epitope.

Reviewer #3 (Remarks to the Author):

The authors have responded very well to the reviewers critiques and comments. The inclusion of the stability assay and showing that P5 is important for the stability of the complex is a key piece of data needed to support their model and conclusions.

Thank you for adding a sentence to the discussion addressing point 3 of Reviewer #2. However, to further nuance the paper and finding, I think it is also important to acknowledge in the title and/or abstract that this change was only found in BA.1 and BA.2 Omicron and NOT BA.4 and BA.5.

Response: We thank the reviewer for this suggestions and have modified the title to reflect the mutation affects early Omicron variants BA.1 to BA.3.